# Genotype I Newcastle Disease Virus, Isolated from Wild Duck, Can Protect Chickens Against Newcastle Disease Caused by Genotype VII

**DOI:** 10.3390/pathogens14040380

**Published:** 2025-04-14

**Authors:** Elizaveta Boravleva, Anastasia Treshchalina, Daria Gordeeva, Alexandra Gambaryan, Alla Belyakova, Irina Gafarova, Alexey Prilipov, Galina Sadykova, Simone Adams, Tatiana Timofeeva, Natalia Lomakina

**Affiliations:** 1Chumakov Federal Scientific Center for the Research and Development of Immune-and-Biological Products, Village of Institute of Poliomyelitis, Settlement “Moskovskiy”, 108819 Moscow, Russia; 2The Scriabin and Kovalenko Federal Scientific Center for All-Russian Scientific Research Institute of Experimental Veterinary, The Russian Academy of Sciences, 109428 Moscow, Russia; 3The Gamaleya National Center of Epidemiology and Microbiology of the Russian Ministry of Health, 123098 Moscow, Russia; 4Institute of Microbiology, Lausanne University Hospital, 1011 Lausanne, Switzerland

**Keywords:** avian orthoavulavirus 1, AOAV-1, Newcastle disease virus, NDV, paramyxovirus, pathogenesis, genotype VII NDV, immunity

## Abstract

Newcastle disease viruses (NDVs) circulating among wild birds and poultry may differ in virulence. Some NDVs cause devastating outbreaks in chickens. The NDV/duck/Moscow/3639/2008 (d3639) strain was isolated from a wild duck. Its genome was sequenced (PP795281, GenBank) and the biological properties, specifically for infection in chicken and mice, were studied. Strain d3639 of genotype I.2 has an F protein cleavage site (112-GKQGRL-117) and a HN protein length (616 a.a.) of the lentogenic pathotype. It was tested, in comparison with the genotype II LaSota vaccine strain, for its immunogenicity and protective efficacy against a challenge with the velogenic NDV strain NDV/chicken/Moscow/6081/2022 (ch6081) of sub-genotype VII.1.1, the complete genome of which was also sequenced in this study (PP766718, GenBank). Both the d3639 and LaSota viruses did not induce clinical signs in chickens or mice. Single immunization was performed by inoculation through drinking water with the live virus. Inoculation protected the chickens during a subsequent challenge with velogenic ch6081 and significantly reduced shedding in feces. Double immunization was sufficient to achieve prolonged immunity and prevented the shedding of the velogenic virus after the challenge. Thus, this natural lentogenic d3639 virus possesses properties similar to the LaSota vaccine strain and can protect against sub-genotype VII.1.1 NDV.

## 1. Introduction

According to current taxonomy [1], the subfamily *Avulavirinae* of the family *Paramyxoviridae* contains the following three genera: *Metaavulavirus*, *Orthoavulavirus*, and *Paraavulavirus*, which include enveloped negative-strand RNA viruses which affect birds. Among the 11 species of the genus *Orthoavulavirus*, the species *Orthoavulavirus javaense* (previous name *Avian orthoavulavirus 1*) represents an etiological agent of the most devastating Newcastle disease (ND) in poultry, causing great economic losses worldwide [2,3]. This agent is commonly known as Newcastle disease virus (NDV) or avian paramyxovirus 1 (APMV-1). The primary hosts of NDV are wild waterfowl; in these hosts, the virus causes asymptomatic infection [4,5]. However, some evolutionary lineages have adapted and acquired virulence in chickens [6]. The NDV strains differ in their pathogenicity in chickens. They may be velogenic (highly pathogenic), mesogenic (with respiratory or nervous signs and a low to moderate mortality), lentogenic (with mild or subclinical respiratory infection), or apathogenic (with subclinical enteric infection) [7]. Since numerous devastating outbreaks of ND in poultry have occurred [6], this disease is listed in the World Organization for Animal Health (WOAH) [8]. Infected birds shed the virus in feces and respiratory secretions. The virus can also be spread through contact with the excreta of infected birds, such as in contaminated feed and water.

Although the main NDV hosts are birds, these viruses can also infect some mammalian species, including humans and small animals, either naturally or experimentally, resulting in either asymptomatic infection or infection with clinical symptoms [9].

Two surface viral glycoproteins, F (fusion) and HN (hemagglutinin-neuraminidase), are responsible for NDV immunogenicity and neutralizing antibody induction in infected organisms. Furthermore, the structure of the F protein cleavage site is a molecular determinant of NDV pathogenicity [4,7].

The broad circulation of NDV in poultry and wild birds has led to a significant genetic diversity of the virus and the emergence of NDV variants, therefore, an updated NDV molecular classification system has been developed on the basis of phylogenetic analyses of the complete F gene sequences [10]. All NDV strains are divided into classes I and II, which are further clustered into genotypes and sub-genotypes according to the topology of the F gene phylogenetic tree [10].

Vaccination provides effective protection against the spread of NDVs. In most countries, including the Russian Federation, the vaccination of industrial poultry against NDV is mandatory [11,12]. The most widely used live vaccine against ND was developed on the LaSota strain in the 1950s [13]. Inoculation with LaSota protects against a wide range of NDVs [14]. However, the emergence of new pathogenic NDV variants during virus circulation may lead to antigenic mismatch between the actual strain and vaccine strain [15,16,17,18]. As a result, outbreaks of NDV continue to be recorded.

In recent years, velogenic NDVs of genotype VII (class II) have become the most widespread in Eurasia and Africa [19,20,21,22]. In the Russian Federation, viruses of this genotype have been circulating since 2006. During subsequent years, NDV sub-genotype VII.1.1 was the cause of sporadic outbreaks among poultry in backyards and private farms and among unvaccinated livestock in different regions [19,23]. Antibodies against NDV were detected in crows, pigeons, and wild birds in four regions of the Russian Federation in 2019. Positive samples from pigeons and other synanthropic birds were obtained near poultry farms. These birds could pose a threat of infection in commercial flocks [24].

Different avian paramyxoviruses have been isolated from wild waterfowl [25,26]. Class II Newcastle disease viruses of sub-genotypes I.2 and VII.1.1 (*Orthoavulavirus*), APMV-4 (*Paraavulavirus*) and APMV-6 (*Metaavulavirus*), were discovered in Dagestan in 2017–2020 [25].

Every year from August to November since 2006, we have been monitoring the presence of avian influenza viruses in ponds and lakes of the Moscow region, where a high density of wild ducks is present. Occasionally, paramyxoviruses were isolated from the collected mallard feces. They had not been studied in detail until a pathogenic NDV was isolated from dead chickens in the backyard of a private house in the Moscow Region in 2022 [19]. The pathogenic NDV/chicken/Moscow/6081/2022 strain was studied in our previous work and was classified as sub-genotype VII.1.1 of class II [19]. In total, 18 avulaviruses were isolated from mallard feces and identified as belonging to *Orthoavulavirus* (NDVs) or *Paraavulavirus* [26]. Since we did not encounter pathogenic paramyxoviruses, even over a long period of virus monitoring, we hypothesized that non-pathogenic duck viruses can protect against a pathogenic virus, in nature. To test our hypothesis, the experiments performed here were conducted in a manner close to natural conditions. Chickens were chosen as the host, as they are NDV-susceptible birds and were infected here through their drinking water.

The purpose of this work was to study the genetic and biological properties of NDV strains isolated from wild ducks in the Moscow region and to analyze their protective abilities against velogenic viruses of sub-genotype VII.1.1, in comparison with the LaSota vaccine strain.

## 2. Materials and Methods

### 2.1. Viruses

The NDV/duck/Moscow/3639/2008 (d3639) virus was isolated from the feces of wild ducks sampled at the side of a pond in Moscow in 2008 (10 August 2008). The pathogenic NDV/chicken/Moscow/6081/2022 (ch6081) virus was isolated (31 August 2022) from a chicken, as described earlier [19]. The viruses were propagated in the allantoic cavity of pathogen-free, 10-day-old embryonated chicken eggs (ECEs) at 36 °C and harvested at 48 h post infection (p.i.). The 50% embryo infectious dose (EID_50_) in the samples was determined by titration in the ECEs. Infectious allantoic fluids (IAFs) were pooled, divided into aliquots, and stored at −70 °C.

The LaSota vaccine strain was kindly provided by Dr. S.G. Markushin, Federal State Budgetary Scientific Institution (The Mechnikov Research Institute of Vaccines and Sera), Moscow, Russian Federation.

### 2.2. Isolation of Avian Viruses

Fresh feces samples were collected on the shores of city ponds in places with a high density of mallards (Anas platyrhynchos) during August–November. The samples were suspended in a double volume of phosphate-buffered saline (PBS) supplemented with 2% MycoKill AB solution (PAA Laboratories GmbH, Pasching, Austria), 0.1 mg/mL of kanamycin, 0.01 mg/mL of nystatin, and 0.4 mg/mL of gentamicin. In the case of virus isolation from dead birds, lungs and kidneys were used according to published standards [7]. In total, 0.5 g of tissue samples was homogenized with glass beads and 2 mL of the antibiotic solution was added. The suspension was clarified by centrifugation at 4000 rpm for 5 min, then 0.2 mL of the supernatants was inoculated into the allantoic cavity of the ECEs. Allantoic fluid was collected after 48 h and tested by hemagglutinin agglutination assay with 1% chicken red blood cells, in accordance with the published protocol [7]. Positive samples were taken for further passaging. Isolated strains were stored in the virus repository of the Chumakov scientific center (Moscow, Russia).

### 2.3. Animals

Chicken embryos and chicks were purchased from poultry farms with special considerations to ensure that the embryos and birds were obtained from laying hens that had not been vaccinated against NDV and had been tested for the absence of pathogens.

The 10-day-old ECEs were purchased from the Ptichnoye State Poultry Farm, Moscow, Russia. ECEs were used for the propagation of viruses. Leghorn chickens aged 3, 15, and 70 days were obtained from a poultry farm (Tomilinskaya, Moscow, Russia) free from infectious avian diseases. Chickens were used to assess the immune responses and pathogenicity/safety of the viruses. BALB/c mice were from Lesnoye farm, Moscow, Russia. Mice were used for the analysis of viral pathogenicity.

### 2.4. Sequencing and Genotyping

Total RNA was isolated from 200 μL of virus-containing allantoic fluid by the guanidinium thiocyanate method [27]. Specific primers were used for reverse transcription and PCR in a single tube by the RT-pFusion kit (Alpha-Ferment, Moscow, Russia). The primer sequences are available on request. The reaction was performed in volume of 20 μL in a thermocycler Gene Explorer (BIOER Technology Co., LTD, Hangzhou, Zhejiang, China), as follows: (1) 42°—40 min, 85°—2 min; (2) 98°—15 s, 50°—15 s, 72°—3 min, repeated 35 times; and (3) 72°—5 min. The amplified fragments were analyzed by electrophoresis in a 2% agarose gel, 50 mM TAE buffer. The fragments were then extracted from the agarose gel using the Diatom DNA Elution kit (D1031, Isogene Laboratory Ltd., Moscow, Russia). Sanger sequencing was carried out using the BrightDye™ Terminator Cycle Sequencing Kit v3.1 (NimaGen, Nijmegen, The Netherlands), and the following analysis was performed using the Genetic Analyzer 3130XL (Applied Biosystems, Foster City, CA, USA). Lasergene Molecular Biology Sequence Analysis Software, v. 6 (DNASTAR Inc., Madison, WI, USA) was used to analyze sequences and generate complete genomes.

The nucleotide sequences determined in the present study were submitted to GenBank under the accession numbers PP795281 for d3639, PP766718 for ch6081, and PP780192 for d3604. The F and HN gene sequences of the LaSota strain which was used in this work coincided with PQ106774 (GenBank).

For genotyping, the complete nucleotide sequences for the coding region of the F gene of class II NDV isolates were downloaded from the GenBank of the National Center for Biotechnology Information (https://www.ncbi.nlm.nih.gov/genbank/ (accessed on 20 April 2024)). Sequence processing was performed using BioEdit 7.2. (https://bioedit.software.informer.com/ (accessed on 20 April 2024)) and MEGA X, v.10.2.6 (https://www.megasoftware.net/ (accessed on 20 April 2024)). All nucleotide sequences were aligned using the MUSCULE algorithm and cut in the reading frame. Maximum-likelihood trees, based on the general time-reversible (GTR) model, were constructed by using the BEAST software package (1.10.4). Analysis was run for over 10,000,000 generations and trees were sampled every 1000 generations, resulting in 10,000 trees. The iTOL v6 online service (https://itol.embl.de/ (accessed on 20 April 2024)) was used to visualize and annotate the trees. Genotypes were determined based on phylogenetic topology according to [10].

### 2.5. Mean Death Time (MDT) Assays

For MDT assays, 10-fold serial dilutions of fresh IAF in PBS were performed. Five ECEs were inoculated with the same dilution in a volume of 0.1 mL per each ECE and were subsequently incubated at 36 °C. The eggs were observed 3 times a day for 5 days and any embryo deaths were recorded. The highest virus dilution that caused 100% mortality was considered as the minimum lethal dose. The MDT was the mean time in hours for the minimal lethal dose to kill the inoculated embryos [28].

### 2.6. Analysis of the Viral Pathogenicity in Chickens

Chickens of the same age and weight were divided into equal groups of 3–4 birds per unit, which were kept in separated cages of different rooms depending on the experimental purpose. To analyze the viral pathogenicity in chickens, the birds were first deprived of water overnight. The following day, 10 mL of water containing 10^8^ EID_50_ of the tested viruses was added to drinking bowls, and one bowl was placed in a cage with 4 birds in the morning. The control group received plain water. The doses were selected empirically in previous studies. The survival and body weight of each chicken were monitored daily. Blood samples were taken before infection (randomly as a control) and on day 14 after infection (from each bird).

### 2.7. Oral Immunization and Challenge with Highly Pathogenic NDV

The chickens were divided into groups of 4 birds per cage and deprived of water overnight. A drinking bowl with 12 mL of a virus solution at a concentration of 10^8^ EID_50_/mL was placed in each cage in the morning. An hour later, the drinking bowls were removed and replaced with bowls with clean water. Each chicken received nearly 10^8^ EID_50_ of the virus. Blood samples were taken on day 14 after the infection. For the double immunization experiment, secondary immunization was conducted as the first infection, 99 days later.

Challenge with ch6081 virus was also performed orally, through drinking water, as described above. The infectious dose was ~10^7^ EID_50_ in 3 mL of solution per chicken. Blood samples were taken on day 14 after the challenge. The experiments were carried out in a biosafety level 3 containment facility.

### 2.8. Assessment of Viral Shedding in Chicken Feces

Following infection, the birds’ feces were sampled once daily. Clean matting was placed inside each cage and all fresh fecal samples were collected within a 30 min timeframe. The samples were suspended in PBS, supplemented with antibiotics, and centrifuged, as previously described (see Section 2.2). The supernatant in a volume of 0.1 mL was then inoculated into 10-day-old ECEs to detect the viable virus. After incubation for 48 h at 4 °C, the allantoic fluids were collected and tested by hemagglutination assay with a 1% solution of chicken red blood cells, in accordance with recommendations [7].

### 2.9. Assessment of Antibody Titers

Blood samples were taken before infection (randomly as a control) and on day 14 after infection (from each bird). The levels of antibodies (ABs) in the chicken sera were assessed by ELISA, as described earlier [29]. Briefly, the 96-well plates were coated with purified d3639 virus and then blocked with 0.2% BSA solution in PBS for 1 h. The blocking solution was removed, and ten-fold serial dilutions of the serum samples were prepared in each row of the plate in the wells in 100 μL of buffer (0.1% Tween-20, 0.2% BSA on PBS), starting from 1:50. Blank wells were those not coated in virus, and sera from uninfected birds served as negative controls. Incubation was performed for 4 h at 4 °C. After washing, peroxidase-labeled antibodies against chicken immunoglobulins (Sigma-Aldrich, Inc., St. Louis, MO, USA) were added into 100 μL of the same buffer. These were incubated for 2 h at 4 °C prior to washing. To develop the plate, a color reaction with tetramethylbenzidine substrate solution was carried out and stop solution was added prior to reading at 450 nm on a microplate reader (Multiskan FC, Thermo Fisher Scientific, Shanghai, China). The presence of antibody binding was determined by the optical density of a specific color. The result is expressed as a titer with ten-fold dilutions of serum.

### 2.10. Analysis of Viral Pathogenicity for Mice

Groups of 8 BALB/c mice were inoculated intranasally, either with a placebo (control) or with diluted IAF in a volume of 50 µL (10^6^ EID_50_ per mouse). The doses used here were selected from previous empirical studies. The survival and body weight of the mice were monitored daily.

### 2.11. Statistics

The arithmetic means, geometric means, standard deviations, and *p*-values were determined by Microsoft Office Excel software 365 or GraphPad Prism v10.4.1. Significant differences between different groups were determined by the Student’s *t*-test, multiple unpaired *t*-tests, or two-way-ANOVA. Values of *p* < 0.05 were considered as significant.

### 2.12. Ethical Considerations

All tests were carried out in compliance with the standard for the keeping and care of laboratory animals GOST 33215-2014, adopted by the Interstate Council for Standardization, Metrology and Certification, as well as in accordance with the requirements of Directive 2010/63/EU of the European Parliament and of the Council of the European Union of 22.09.2010 on the protection of animals used for scientific purposes. All necessary measures were taken to alleviate the suffering of the animals. The study design was approved by the Ethics Committee of the Chumakov Federal Scientific Center for the Research and Development of Immune-and-Biological Products, Village of Institute of Poliomyelitis, Settlement “Moskovskiy”, 108819 Moscow, Russia (Approval #4 from 7 December 2019).

## 3. Results

During the monitoring of wild ducks flying through Moscow, 28 avulaviruses were isolated from mallard feces [26]. Five NDV viruses were identified by RT-PCR, and all were apathogenic in chickens. One of these viruses, NDV/duck/Moscow/3639/2008 (d3639), was studied in this work. To define their genotype and virulence and to estimate the epitope matching between the three studied strains (d3639, LaSota, and ch6081), sequencing of the viral surface genes was performed. Pathogenicity was determined in both chickens and mice. Immunogenicity and potential protectivity against velogenic sub-genotype VII.1.1 were tested in chickens in comparison with the LaSota vaccine strain.

### 3.1. Sequencing and Genotyping of Moscow NDV Isolates

The complete genomes of the previously identified d3639 and ch6081 viruses, as well as the partial genomes of NDV/duck/Moscow/3604/2008 (d3604) and the LaSota virus (F and HN genes), were sequenced. The sequences were submitted to GenBank (Section 2.4).

The genotyping of the d3639, ch6081, and LaSota strains was performed according to [10]. The coding region of the F gene was used for phylogenetic analysis. The data set contained two representatives from each genotype of NDV class II (*n* = 125) and also the nearest relatives of the Moscow isolates, which were determined by BLAST v.1.3 (https://blast.ncbi.nlm.nih.gov/Blast.cgi (accessed on 17 March 2025)). It was found that the ch6081 isolate belongs to sub-genotype VII.1.1, while the d3639 and d3604 viruses belong to sub-genotype I.2 (Figure 1).

These results indicate that the strains analyzed in this study belong to the following three different genotypes of NDV class II: lentogenic d3639 belongs to genotype I (sub-genotype I.2), the vaccine LaSota belongs to genotype II, and pathogenic ch6081 belongs to genotype VII (sub-genotype VII.1.1) (Figure 1).

### 3.2. Fusion (F) and Hemagglutinin-Neuraminidase (HN) Protein Sequence Comparison Between d3639, LaSota, and ch6081

The amino acid sequences of the surface glycoproteins F and HN were compared between the lentogenic d3639, vaccine LaSota, and pathogenic ch6081 NDV strains of genotypes I, II, and VII, respectively.

The main feature which determines NDV pathogenicity is the structure of the F protein cleavage site [4,7]. Two strains, d3639 and LaSota, have the 112-GKQGRL-117 motif, which is typical for avirulent and lentogenic viruses. Contrarily, the ch6081 strain has the 112-RKQKRF-117 motif, which indicates that this strain has velogenic properties (Figure 2).

A comparison of the F proteins of the three studied NDV strains revealed 9–11 mismatches in the signal peptide (1–31 a.a.) relative to the d3639 strain. Additionally, 2–3 substitutions (C514F, I516A, and V520G, relative to d3639 strain) were found between ch6081 and the lentogenic strains in the transmembrane region (500–522 a.a.), as well as in the adjacent region 482–489 (E482A, N485K, and D489E). Finally, within the fusion peptide (117–142 a.a.), which may aid the function of fusion [30], two substitutions (G124S and S139A) are included in ch6081 (relative to the d3639 strain).

As the existence of N-glycosylated amino acid residues near the epitope can affect the antigenic properties of a virus, we determined the locations of the N-glycosylation sites in the surface proteins of the tested NDV strains by utilizing the on-line program NetNGlyc 1.0 [31]. All studied strains have six predicted N-glycosylation sites in positions 85, 191, 366, 447, 471, and 541. There is only a single difference at T385A in the predicted short linear epitope 379–387 between d3639 and ch6081. The d3639 and LaSota strains differ by two substitutions at T203A and A442V in the antigenic epitopes, corresponding with 198–206 and 436–444.

**Figure 2 pathogens-14-00380-f002:**
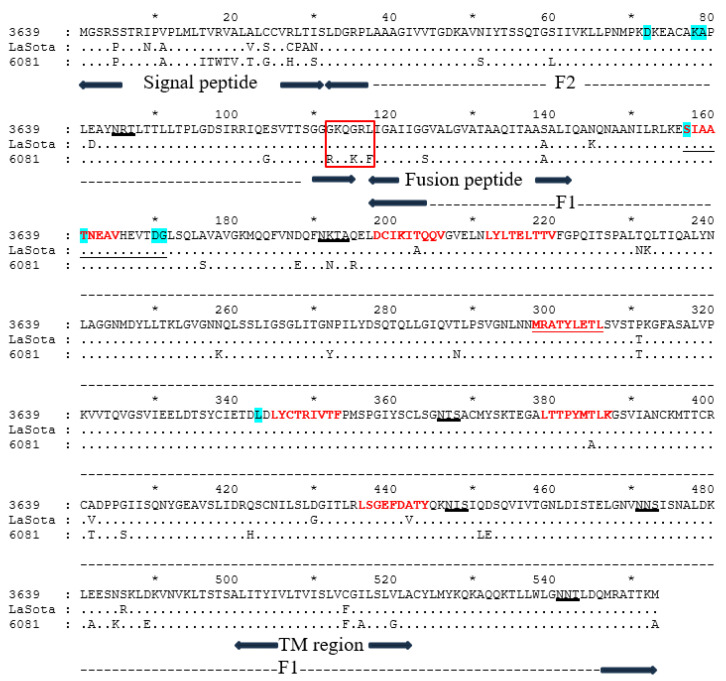
Amino acid sequences and schematic structure of the F protein [32]. The boundaries of the F2 (32–116 a.a.) and F1 (117–553 a.a.) subunits, signal peptide (1–31 a.a.), fusion peptide (117–142 a.a.), and transmembrane region (TM, 500–522 a.a.) are designated by arrows. The motif at 112–117 in the F2/F1 cleavage site is boxed. Predicted N-glycosylation motifs are underlined in black. Predicted short liner epitopes are shown in red font [33]. Epitopes determined by neutralizing monoclonal antibodies [34] are highlighted in blue.

Alignment of the HN proteins of the three NDV strains (d3639, LaSota, and ch6081) revealed a variance in the length at their C-terminus (Figure 3). Thus, the sizes of the complete proteins are 616 a.a. in d3639, 577 a.a. in LaSota, and 571 a.a. in ch6081. As was shown earlier, an elongated HN C-terminus did not affect the tropism of the virus, but did decrease its pathogenicity [35]. In contrast, a virus possessing a truncated HN was more pathogenic [36]. A study of the conformational structure proposed that an elongated C-terminus may block the binding site to the sialic acid cellular receptor [36].

Most of the differences between the lentogenic (d3639 and LaSota) and velogenic (ch6081) strains were observed at the N-terminal region (1–125 a.a.) of HN. This region coincides with the transmembrane and stalk regions of the protein.

Five predicted N-glycosylated motifs (119, 341, 433, 481, and 538) were found in the LaSota vaccine strain. An additional motif (600 a.a.) is present in the elongated C-terminus of the d3639 strain. However, the predicted site at 538 does not glycosylate due to its location in a fragment (531-CFKVVKTNKTYC-542) flanked by two cysteines which form a disulfide bond [41]. The strain ch6081 has also lost this site (538 a.a.) due to a T540V substitution. In addition, ch6081 has a predicted N-glycosylated site at 144 a.a., in addition to the previously mentioned sites 119, 341, 433, and 481.

The HN protein contains four short linear and three conformational epitopes (cited according to [17], Figure 3). The structures of the two linear epitopes (131–139 and 193–201) of the LaSota vaccine strain and the velogenic ch6081 strain are in accordance but differ from d3639 at E134G and R201H. The LaSota and d3639 strains used for vaccination have an identical structure for the two linear epitopes (242–256 and 345–353) and two conformational epitopes (263, 287, and 321 a.a. and 332, 337, and 356 a.a.), although they have one difference at D494G in the third conformational epitope (494, 513–521, and 569 a.a.).

The velogenic ch6081 strain differs from the lentogenic strains d3639 and LaSota by E256G, N263K, E347Q, K356E, and I514V substitutions, which are located in the five predicted antigenic epitopes.

The percent identities of the F and HN proteins of three studied strains were determined (Table 1). The lentogenic LaSota and d3639 strains have a 93.3% similarity in their F proteins. Even though their HN proteins differ in length, they have a 94.9% identity in the overlapping region (1–571 a.a.). The similarity of the F and HN proteins between the velogenic ch6081 virus and the lentogenic strains ranges from 86.6 to 89.2%.

### 3.3. Pathotypes of d3639 and ch6081 Viruses

Our previous study showed that the MDT of the ch6081 velogenic virus was 52 h [19]. Here, the MDT was measured in the same way as in our previous study, and for the d3639 virus it was more than 90 h, which indicates the lentogenicity of this strain [28].

### 3.4. Pathogenicity of d3639 for Chickens

The bird infection experiments were performed in a manner which was similar to natural exposure in water. Therefore, the virus was administrated through drinking water. The doses were experimentally estimated previously. In these experiments, the doses used were clearly higher than those in nature.

The high pathogenicity of the ch6081 strain for chickens has been shown previously [19]. Here, we evaluated the pathogenicity of the d3639 virus. Chicken groups (*n* = 6 in group) of different ages were given 10^8^ EID_50_ of the d3639 virus in their drinking bowls. No signs of disease were observed for 3-day-old, 15-day-old, and 70-day-old chickens after infection. The weight gain curve for 3-day-old chickens infected with the d3639 virus follows the curve for the control chickens (Figure 4). Numerous experiments were performed with the d3639 virus, but the mortality rate never exceeded the mortality in the control group. The highly pathogenic ch6081 virus, however, was lethal during all infection schemes for non-immune chickens.

### 3.5. Pathogenicity of d3639 and ch6081 for Mice

To compare the pathogenicity of d3639 and ch6081 in mice, two groups of eight mice were infected with these viruses intranasally at 10^6^ EID_50_ per mouse. The third group (control) was inoculated with PBS. The survival and weight of the mice were monitored for 14 days following infection. In both infected groups, an initial decrease in weight was observed compared to the control group. However, after the seventh day, the infected mice gained weight, and by day 14, all mice had recovered (Figure 5). Interestingly, the ch6081 virus, which is very pathogenic for chickens, is lentogenic in mice, comparable to the d3639 virus.

### 3.6. Immunogenicity and Protectivity of d3639 and LaSota After Single or Double Oral Administration to Chickens

Antibody titers were measured by ELISA using plates coated with NDV virus. Preliminary experiments comparing these titers using the ch6081 or d3639 viruses showed nearly identical results. Therefore, in further experiments, the plates were coated with the lentogenic d3639 virus.

To compare the immunogenicity of the d3639 virus and the LaSota vaccine strain, the level of anti-NDV IgG in chickens was evaluated on day 14 p.i. The infectious dose for each of the viruses was approximately equal to 10^8^ EID_50_ per chicken. When chickens were infected with either of these viruses at three days of age, the antibody titers were virtually identical between the two viruses. When these animals were further challenged with ch6081, two weeks after the initial infection, the mortality rate of the chickens was 0% and the shedding of the challenge virus in feces was only detected once (Table 2).

Subsequent experiments with double immunization were also performed. The birds were booster-immunized prior to egg laying. Here, the antibody titers were measured 2 and 14 weeks after the first administration of the LaSota or d3639 viruses. During that time, the antibody titers in the chickens dropped by 10-fold. Then, the chickens were orally re-infected, as before. The antibody titers increased by nearly two orders of magnitude (Table 3). All chickens survived following a challenge with the velogenic genotype VII.1.1 ch6081, without any signs of disease. After this challenge, the antibody titers increased by another order of magnitude. Some weeks later, an assessment of the egg production of hens showed a production value equal to the norm. The resulting egg yolks were evaluated for antibodies and contained IgY antibodies against NDV in high titers. We used these antibodies as a reagent in subsequent work [29].

### 3.7. Viral Shedding in the Feces

Viral shedding was determined by bioassay in ECEs, as described in Section 2.8. Chickens infected with the d3639 and LaSota viruses excreted the viruses in their feces from days 3 to 6 p.i. The virus detection rates in these groups ranged from 30% to 80%. Chickens infected with ch6081 also shed the virus prior to death. The chickens that were inoculated once with the d3639 or LaSota viruses experienced a sharp reduction in the shedding of ch6081 after challenge. The chickens that were inoculated on two occasions with d3639 or LaSota did not shed ch6081 after the challenge at all.

## 4. Discussion

NDVs are widely spread around the world and can affect plenty of species of birds, including poultry (chicken and turkey) and wild birds (duck, goose, swan, psittacine, cormorant, anhinga, gull, and others) [5,6]. Since these various species differ in susceptibility, infection with NDVs induces different clinical signs or can be asymptomatic. Poultry, mainly chickens, are more susceptible to infection, while waterfowls, like ducks and geese, were considered to be resistant against NDVs until outbreaks occurred in geese (1997) and ducks (2002) in China. The pathobiology of NDVs in waterfowl was well-described in a review by Rehman et al. [42]. Although wild waterfowl viruses are generally lentogenic or potentially pathogenic, they can play a role in the transmission, spread, and emergence of ND in poultry. Some lentogenic isolates may be converted into pathogenic viruses through serial passaging in susceptible birds [42].

Four major ND panzootics were recorded in the 20th century. In the mid-1940s, the first vaccines against NDV were developed and began to be widely used [13]. Although various commercial and modern recombinant experimental vaccines exist now, the classical live vaccines are still used in many countries [43]. The currently used live and inactivated commercial vaccines for poultry are based on adapted apathogenic or lentogenic strains of genotypes I or II in chickens. However, velogenic NDVs of other genotypes have also emerged in vaccinated flocks. During recent years, the most dangerous velogenic genotype VII has been circulating worldwide and has been isolated among vaccinated birds [15,16,17,18].

Some authors explain these events by a mismatch between the genotype of the vaccine used and that of the field viruses. A genotype-mismatched vaccine may defend birds against clinical signs of disease but not completely from shedding the virus. If a velogenic virus appears in a vaccinated flock, it may result in a devastating disease [15,43,44].

Two surface viral glycoproteins, F and HN, play important roles in the immunogenicity and pathogenicity of NDV. Both proteins possess immunogenic epitopes. Am antigenic distinction in the neutralizing epitopes between the circulating velogenic viruses and the vaccine strain may explain the presence of ineffective vaccination against ND [45]. A high similarity in the F and HN proteins between field and vaccine strains might prevent severe clinical signs and viral shedding [11,15,46,47].

Modern techniques based on reverse genetics were applied to obtain the VII genotype-matched vaccines [48,49]. One of the vaccines was constructed due to an F protein cleavage site modification which is present in the circulating NDV strain of genotype VII in Malaysia [48]. The other was a DNA vaccine with a plasmid construction containing only the F and HN genes from the virulent NDV genotype VII strain isolated in Tanzania [49]. Recently, another vaccine, a chimeric bivalent vaccine based on virus-like particles (VLPs), was developed to protect domestic geese against virulent genotype VII NDV and gosling plague [50]. Notably, all of these vaccines were tested in comparison with the classical LaSota vaccine. All tested vaccines protected birds from mortality against a challenge with genotype VII NDV. The vaccine administration route varied depending on the type of vaccine. The DNA vaccine was administrated via intramuscular injection rather than the preferred route by eye drop [49]. The live attenuated vaccines, LaSota and the Malaysian recombinant strain, were administrated via the occulo-nasal route [48]. The VLP and LaSota vaccines were injected subcutaneously in the necks of geese [50]. It should be noted that the authors of the original LaSota strain recognized that the strain could be vaccinated intranasally, ocularly, or in drinking water, as well as intramuscularly, even from the beginning [13].

In the present work, the similarities of the F and HN proteins between the lentogenic d3639 and velogenic ch6081 strains were 89.2 and 89.8%. These values were slightly higher than the 86.6 and 87.7% between the LaSota and d6081 strains (Table 1).

The high similarity of the analyzed short linear epitopes in the surface proteins (F and HN) can partially explain the immunogenic cross-protection of the lentogenic strains (LaSota and d3639) against pathogenic NDV ch6081, which belong to different genotypes. However, additional study is needed to confirm this hypothesis.

Our results show that the d3639 strain of genotype I is non-pathogenic for chickens and mice but still induces an antibody response. These antibodies are comparable in efficacy to the LaSota strain and can protect against a challenge with the genotype VII.1.1 ch6081 virus, also stopping cloacal shedding.

Similar results were observed in other studies where the LaSota vaccine strain (genotype II) was used to protect against NDV genotype VII [15,17,44]. They revealed that virus shedding in the vaccinated birds was generally limited to within the first 5 days after challenge and was dependent on the level of neutralizing antibodies [15,44].

The live commercial vaccines currently used on poultry farms are from a strain of genotype I or II [43]. Elbestawy et al. [16] compared the properties of two vaccine strains from genotypes I and II. They showed that the vaccine based on genotype I (V4 strain) protected broiler chickens more efficiently than genotype II (LaSota strain) against a challenge with a velogenic genotype VII.1.1 NDV.

Genotype I is mainly composed of avian avulaviruses that have been isolated from wild birds and are mostly apathogenic. The lentogenic d3639 NDV studied here also belonged to genotype I. No distinctions in immunogenic or protective properties were found between the d3639 and LaSota strains even though they belong to two different genotypes, I and II, respectively. When each was utilized for immunization prior to challenge with a velogenic genotype VII.1.1 ch6081 strain (Table 2 and Table 3), both offered protection against disease and viral shedding. In our experiments, the d3639 virus was practically indistinguishable from LaSota in terms of immunogenicity and protectiveness. The comparison of their viral surface glycoprotein sequences (F and HN) did not find any essential differences in their predicted antigenic epitopes and N-glycosylation sites.

Finally, the study showed that the NDV/duck/Moscow/3639/2008 is a lentogenic virus with an MDT of more than 90 h and belongs to the I.1.2 sub-genotype with an F cleavage site structure of the apathogenic phenotype. This strain does not cause clinical signs in chickens or mice at a dose of 10^6^ EID_50_ per individual. The immunogenic and protective properties of the d3639 strain were tested in chickens in comparison with the LaSota vaccine strain. The amino acid sequences of the F and HN surface proteins were analyzed for lentogenic (d3639 and LaSota) and velogenic NDVs. The high match in the predicted short linear epitopes between the tested lentogenic (d3639 and LaSota) and velogenic ch6081 strains can partially explain the immune protection of chickens inoculated with these lentogenic viruses against a sub-genotype VII.1.1 NDV ch6081 challenge.

These characteristics indicate that the d3639 strain could serve as a candidate for creating a live vaccine. Even so, additional research is needed to assess all parameters which a viable vaccine strain must possess.

## 5. Conclusions

This study shows that the wild duck virus d3639 matches LaSota in protecting chickens and can prevent the circulation of HP genotype VII NDV. This strain represents a promising live vaccine candidate for the protection of poultry against NDV via oral vaccination. Future directions of research will be to study the d3639 strain in more detail as a vaccine candidate. The main purpose of the best vaccine candidate is both to prevent against mortality and severe clinical signs, as well as to stop viral shedding.

## Figures and Tables

**Figure 1 pathogens-14-00380-f001:**
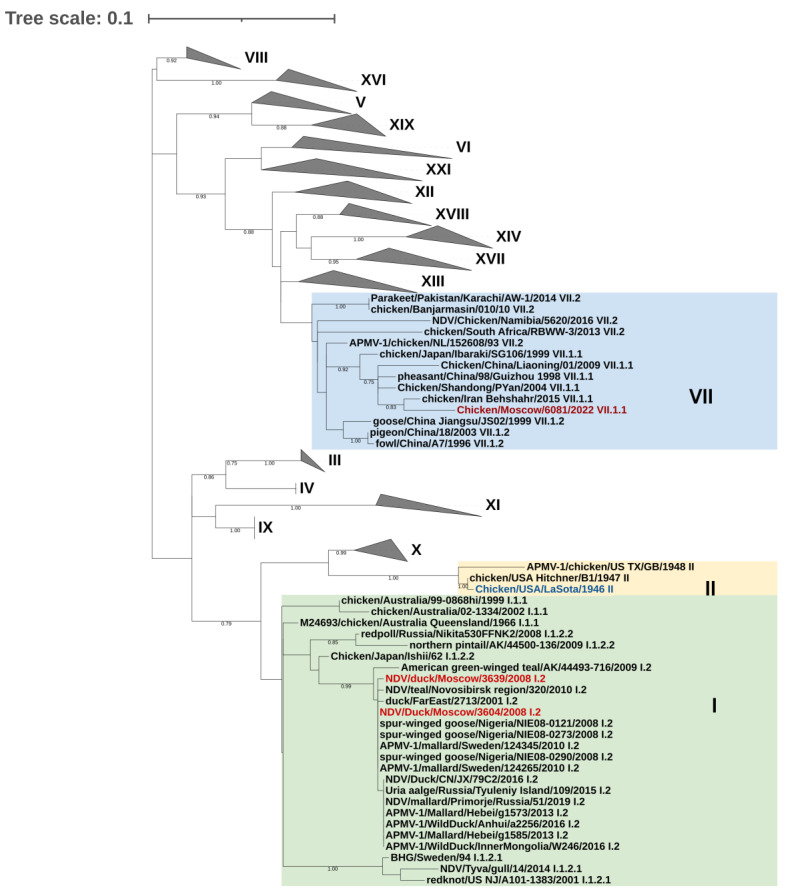
Genotyping of d3639, ch6081, and LaSota strains based on the F gene phylogenetic tree topology suggested by [10] for NDV class II. The strains from the Moscow region are marked in red. Viruses of genotypes not associated with the study are grouped (grey triangles). Genotypes are designated in Roman numerals. Sub-genotype is indicated near strain name.

**Figure 3 pathogens-14-00380-f003:**
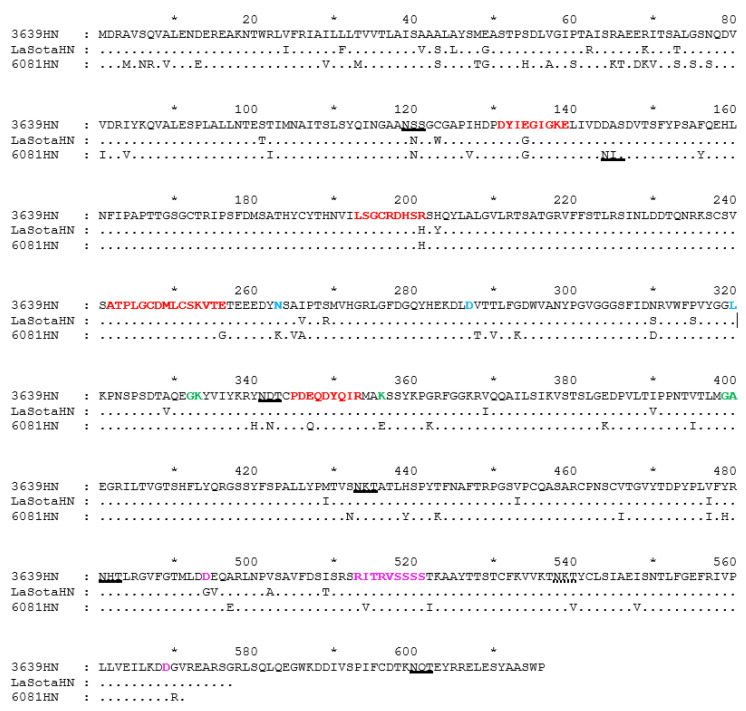
The HN protein structure of the lentogenic strains d3639 and LaSota and the velogenic ch6081. The epitopes are designated based on the data of Chang et al. [17]. Short linear epitopes (193–201 a.a.), (242–256 a.a.), (345–353 a.a.), and (131–139 a.a.) are highlighted in red. Conformational epitopes (263, 287, 321 a.a.), (332, 337, 356 a.a.), and (494, 513–521, 569 a.a.) are highlighted in blue, green, and light violet, respectively. Predicted N-glycosylation motifs are underlined in black. The non-N-glycosylated motif (539–541 a.a.) is designated by a broken line. The HN epitopes were determined earlier by [17,37,38,39,40].

**Figure 4 pathogens-14-00380-f004:**
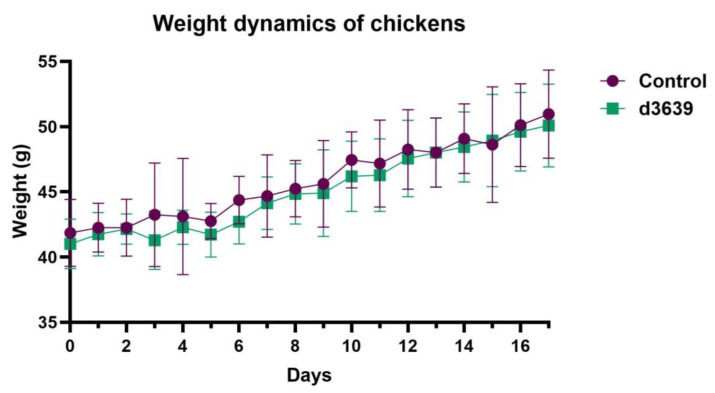
Weight dynamics of 3-day-old chickens infected with NDV/duck/Moscow/3639/2008 (d3639) strain in comparison with non-immune (control) chickens. Weight in grams is represented as arithmetic mean for each group of animals (*n* = 6). No significant differences according to multiple unpaired *t*-tests.

**Figure 5 pathogens-14-00380-f005:**
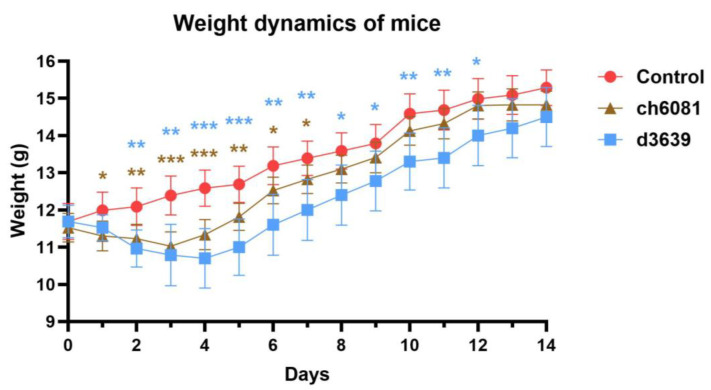
Weight dynamics of mice infected with strains NDV/duck/Moscow/3639/2008 (d3639), NDV/chicken/Moscow/6081/2022 (ch6081) and phosphate-buffered saline (control). Weight in grams is represented as arithmetic mean for each group of animals (*n* = 8). Initially, there was a significant decrease in the weight of mice following inoculation with ch6081 or d3639 compared to control mice, but by day 13, both groups recovered relative to the control. Significance was determined by two-way-ANOVA between each group and the control, per day, where (*) *p* < 0.05, (**) *p* < 0.01, and (***) *p* < 0.001.

**Table 1 pathogens-14-00380-t001:** Pairwise amino acid sequence comparison of F and HN proteins between different NDV strains (percent identity).

	F Protein		HN Protein (Part 1–571)	
Strain	LaSota	ch6081	LaSota	ch6081
d3639	93.3	89.2	94.9	89.8
LaSota		86.6		87.7

**Table 2 pathogens-14-00380-t002:** Post-infection and post-challenge survival, excretion of the viruses in feces, and levels of antibodies against NDV in chicken sera after single immunization.

		Age (Days) of Chickens					
		3	6–11	18	20	23	34
Gr	N *	Infection	Shedding **	AB ***	Challenge	Shedding	Survival
G1	7	LaSota	7/12	3.3 ± 0.4	ch6081	1/15	7/7
G2	7	d3639	8/12	3.4 ± 0.3	ch6081	0/16	7/7
G3	7	Control	-	<1.6	ch6081	6/6	0/7

Notes: Chickens were infected or challenged with indicated viruses by oral administration through water drinking bowls. Designations: Gr—group number. N *—the number of chickens in the group. Shedding **—excretion of the virus in the feces on days 3 to 6 after infection or challenge. Number of virus positive/total number of fecal samples. AB ***—Antibody titers are expressed as the decimal logarithm of the geometric mean value and standard deviation. The antibodies against NDV were determined by ELISA. Survival—surviving/total number of birds used for primary virus administration.

**Table 3 pathogens-14-00380-t003:** Pathogenicity, virus shedding in feces, and antibody level in chickens after double immunization.

		Age (Days) of Chickens								
		3	18	100	102	116	120	123–128	134	134
Gr	N	Primary Infection	AB	AB	Secondary Infection	AB	Challenge	Shedding	Survival	AB
G4	8	LaSota	3.5 ± 0.2	2.3 ± 0.6	LaSota	4.2 ± 0.5	ch6081	0/15	8/8	5.2 ± 0.1
G5	8	LaSota	3.5 ± 0.2	2.3 ± 0.6	d3639	4.3 ± 0.4	ch6081	0/16	8/8	5.1 ± 0.2
G6	8	d3639	3.2 ± 0.4	2.1 ± 0.5	d3639	4.3 ± 0.6	ch6081	0/13	8/8	4.9 ± 0.3
G7	8	Control	<1.6	<1.6	-	<1.6	ch6081	7/9	0/8	-

Notes: The same as for Table 2.

## Data Availability

Data are contained within the article.

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
