# Peer review of "Genotype I Newcastle Disease Virus, Isolated from Wild Duck, Can Protect Chickens Against Newcastle Disease Caused by Genotype VII"

_pathogens, 2025, doi:10.3390/pathogens14040380_

Round 1
Reviewer 1 Report
Comments and Suggestions for Authors
Major comments:
- Lines 421-428: The conclusion that the d3639 strain "could serve as a live vaccine candidate" requires more rigorous discussion of safety concerns, particularly the risk of reversion to virulence. Wild-type isolates, even lentogenic ones, typically undergo extensive attenuation before vaccine development. Address stability during passaging.
- Lines 373-378: The viral shedding analysis lacks quantification. Merely reporting presence/absence of virus in feces is insufficient for vaccine evaluation. Quantitative RT-PCR data showing viral load reduction would significantly strengthen your findings and allow proper comparison between vaccination groups.
- Lines 219-227 and Figure 1: The phylogenetic analysis includes relatively few reference sequences for genotype VII.1.1, which is problematic given that this is the challenge strain's genotype. A more robust phylogenetic framework including contemporary circulating strains would strengthen the relevance of your findings.
- The study design section (Lines 92-123): The rationale for choosing the specific infectious doses (108 EID50 for immunization, 107 EID50 for challenge) is not explained. Were dose-response studies conducted previously? How do these doses compare to field exposure levels?
- Lines 337-342: The double immunization protocol demonstrates strong protection, but the interval between primary and secondary immunization (approximately 100 days) is quite long. A discussion of whether shorter intervals would be equally effective would enhance the practical application of your findings.
- Lines 403-406: The immunization studies would benefit from larger sample sizes. The relatively small number of chickens per group (n=7-8) limits the statistical power of your conclusions. This limitation should be acknowledged.
- Lines 340-367: The study doesn't address the critical issue of maternal antibody interference, which is essential for practical application in commercial settings where breeding flocks are typically vaccinated.
- Lines 399-402: The mechanism of cross-protection against heterologous genotypes is insufficiently explored. While epitope analysis is presented, functional studies (such as cross-neutralization assays) are missing, limiting the mechanistic understanding of the observed protection.
Minor comments:
- Lines 42-55: In the introduction, consider providing more context about previous attempts to use genotype-matched vaccines against field strains of NDV.
- Lines 317-335 and Figures 4-5: The figures showing weight dynamics would be more informative with error bars to indicate variation within the groups.
- Lines 245-252: The analysis of epitope differences between strains is thorough, but it would be helpful to explicitly link these differences to potential implications for cross-protection.
- Lines 178-189: Some methodological details regarding the ELISA assay could be expanded, particularly regarding the quantification of antibody titers.
- Lines 425-428: The manuscript lacks economic analysis comparing the potential production costs of a d3639-based vaccine versus existing vaccines. Given the suggestion of the strain as a vaccine candidate, addressing production feasibility is relevant.
- Lines 94-100: The authors should address the risk of using a wild bird isolate that might contain other pathogens or genetic elements not detected in sequencing. Standard vaccine development includes extensive safety testing beyond what is presented.
- Lines 230-239: Consider expanding the comparison between the F and HN protein sequences to include more specific details about functional domains that might influence cross-protection.
Reviewer 2 Report
Comments and Suggestions for Authors
The manuscript presents new information regarding the defence process against Newcastle disease of ducks
- Please describe in greater detail the defence response to the virus and in particular the cross-immunity between genotypes.
- Please explain the gaps in the international literature that would be filled from this work. Please highlight the advantages offered by the current findings in comparison to previous results by other workers.
- 3. Please add a new sub-section in M&M to describe in great clarity all the control procedures and material employed in this study.
- Please describe the season the year when samples were collected and please explain if there were seasonal differences in the findings.
- Did you carry out Bonferroni adjustments in the handling of results for analysis? If not, please justify. Also, perhaps you can carry out a multivariable analysis to detect sources of variance in the findings.
- The challenge procedure must described in greater detail.
- The visualization of the manuscript is not adequate. Please increase the use of tables and graphs to present the results and also reduce the relevant text.
- Some relevant references describing work in geese can be useful in the discussion of results and I suggest to include in the revised manuscript.
- The Discussion is short and does not fully explain the findings. Please extend and please include examples from other species.
- The conclusions section does not provide a take-home message. Also, please also explain the future directions of this research.
Overall. Useful study. Extensive changes and re-evaluation after correction.
Reviewer 3 Report
Comments and Suggestions for Authors
In this study, the authors evaluated the pathogenicity and the protective efficacy of NDV isolate "d3639", isolated from wild ducks in Russia during 2008. Its efficcacy was evaluated against velogenic NDV strain "ch6081" isolated from chicken in Moscow in 2022. this was found previously to belong to genotype VII ( sub-genotype 1.1).
Comments:
1 - In this study, the molecular characterization and sequencing revealed that "d3639" NDV strain is lentogenic and belongs to genotype I (sub-genotype 1.2) . we found this part of the work is well done and perfect.
2) We found , the parts related to the pathogenicity and protection efficacy of "d3639" were not performed according to WOAH fo investigating the pathogenicity of ND viruses. In fact the two tests recommended are : ICPI and IVPI and the chickens must be SPF.
3) in this you did use commercial chickens, since ND vaccination is mandatory in Russia , the chicks had antibodies to NDV, which may limit the virulence of "d3639" and reduce the clinical signs.
4)Eliza is not the gold standard test for NDV; the test recommande by WOAH is HI test.
5) While evaluating the protection or the pathogenicity under experimental condition we use the inoculation individually in order each chick had the same amount of virus particles.
6) Figured 4: not correct ,in fact the weight evolution could not be only 10g between Day 1 and 17. At this period for light breed the weight gain is between 6 and 8g /day
Round 2
Reviewer 1 Report
Comments and Suggestions for Authors
The authors have satisfactorily addressed most of the major concerns and have made appropriate revisions to strengthen the manuscript.
Minor Revisions Still Required
A few minor issues should be addressed before final acceptance:
- The authors should ensure consistent formatting of references throughout the manuscript
- Figure 5 caption should be expanded to better explain the significance of the weight dynamics in mice
- A few typographical errors need correction (e.g., line 123, line 347)
Author Response
Dear Reviewer,
Thank you very much for your reviewing and comments.
Revised section and text are highlighted in yellow.
Comments 1: The authors should ensure consistent formatting of references throughout the manuscript
Response 1: The reference list was corrected.
Comments 2: Figure 5 caption should be expanded to better explain the significance of the weight dynamics in mice.
Response 2: The Figure 5 and its caption were changed.
Comments 3: A few typographical errors need correction (e.g., line 123, line 347).
Response 3: Done.
Reviewer 2 Report
Comments and Suggestions for Authors
The manuscript has become fantastic after the changes and it can be published immediately. The work described is extra super excellent first-class study.
Author Response
Dear Reviewer,
Thank you very much for your review and appreciation of our work.
Revised section and text are highlighted in yellow.
Reviewer 3 Report
Comments and Suggestions for Authors
1) oral vaccination and challenge need more information ( volume of water in bowls for each age, duration that viral solution is consumed in each group)
2) add in material and method : How double vaccination was conducted.
3) table 2 and 3 , the results presented of antibody titres are very likely to be related to those obtained with HI test rather than ELISA ?
4) surprisingly after challenge the antibody titres did not show high increases as we usually seen following challenge: Explain why?)
Author Response
Dear Reviewer,
Thank you very much for your reviewing and comments.
Revised section and text are highlighted in yellow.
Comments 1: oral vaccination and challenge need more information ( volume of water in bowls for each age, duration that viral solution is consumed in each group)
Response 1: Done. We revised the section “2.7. Oral immunization and challenge with highly pathogenic NDV” with a detailed procedure.
Comments 2: add in material and method : How double vaccination was conducted.
Response 2: Done. Lines 187-192.
Comments 3: table 2 and 3 , the results presented of antibody titres are very likely to be related to those obtained with HI test rather than ELISA ?
Response 3: No. It was obtained with ELISA. Antibody titers are expressed as the decimal logarithm, therefore titers were up to 100 000.
Comments 4: surprisingly after challenge the antibody titres did not show high increases as we usually seen following challenge: Explain why?
Response 4: The 10-fold dilutions were prepared. According to Table 3, on day 100 (column #5) after the first infection, the antibody titers were about 2.3 lg. On day 116 (column #7), they increased up to about 4.2 lg after the second infection. Thus, the antibody titer increased by 100 times 2 weeks (116 day) and by 1000 times 4.5 weeks (134 day, column #11) later after the second immunization.
Round 3
Reviewer 3 Report
Comments and Suggestions for Authors
I would be more pleased if HI test was used instead of ELISA